# Low-spin state of Fe in Fe-doped NiOOH electrocatalysts

Zheng-Da He [1,2], Rebekka Tesch [1,2,3], Mohammad J. Eslamibidgoli [1,2], Michael H. Eikerling [1,2,3] & Piotr M. Kowalski [1,2] ✉

Doping with Fe boosts the electrocatalytic performance of NiOOH for the oxygen evolution reaction (OER). To understand this effect, we have employed state-of-the-art electronic structure calculations and thermodynamic modeling. Our study reveals that at low concentrations Fe exists in a low-spin state. Only this spin state explains the large solubility limit of Fe and similarity of Fe-O and Ni-O bond lengths measured in the Fe-doped NiOOH phase. The low-spin state renders the surface Fe sites highly active for the OER. The low-to-high spin transition at the Fe concentration of ~25% is consistent with the experimentally determined solubility limit of Fe in NiOOH. The thermo-dynamic overpotentials computed for doped and pure materials, $\eta = 0.42\,\mathrm{V}$ and 0.77 V, agree well with the measured values. Our results indicate a key role of the low-spin state of Fe for the OER activity of Fe-doped NiOOH electrocatalysts.

The large-scale production of green hydrogen via water electrolysis will play a vital role in the future energy landscape[1]. The successful deployment and scale-up of water electrolysis hinges on the availability of highly stable and active electrocatalyst materials. In this context, nickel (oxy)hydroxides ($NiO_xH_y$) are widely investigated as electrocatalysts for the oxygen evolution reaction (OER), which is critical for water splitting[2–4]. These materials exhibit an intriguing phase transformation behavior under electrochemical conditions that are usually displayed as a so-called *"Bode diagram"*[5]. The charging-discharging cycle of a Nickel (oxy)hydroxide involves four different phases: $\beta$-Ni(OH)$_2$, $\beta$-NiOOH, $\gamma$-NiOOH and $\alpha$-Ni(OH)$_2$. $\beta$-Ni(OH)$_2$ that is stable at potentials below 1.3 V vs RHE, deprotonates to $\beta$-NiOOH in the potential range from 1.3 to 1.5 V vs RHE[6]. Upon oxidation, $\beta$-NiOOH, transforms into $\gamma$-NiOOH, which is characterized by the presence of aqueous interlayers. This phase reduces further to the hydrated $\alpha$-Ni(OH)$_2$, and finally to the de-hydrated $\beta$-Ni(OH)$_2$ phase.

The OER activity of Ni oxide or (oxy)hydroxide-based electro-catalysts drastically increases upon doping even with trace amounts of Fe[7], as had been first discovered by Corrigan[8] and then confirmed in numerous follow-up studies[2,9–11]. Friebel et al. found that the electro-catalytic activity of NiOOH increases by about three orders of magnitude upon mixing with 25% of Fe, a concentration that is close to the solubility limit of Fe in NiOOH[7,9,12]. To date a sound mechanistic explanation of the OER activity enhancement upon Fe incorporation has not been found. Interestingly, the steep growth in activity stops abruptly at an Fe concentration of about 25%. At higher concentration of Fe, the OER activity remains nearly constant. It has been speculated that this behavior could be associated with the solubility limit of Fe in the NiOOH phase, the resulting miscibility gap and the co-existence of Ni- and Fe-rich phases[9]. However, whether this de-mixing is driven by the thermodynamics of solid solutions[13] or another mechanism has not been clarified.

Because of the obvious correlation between OER activity and Fe incorporation, most experimental studies have suggested Fe as an active site, e.g.,[9–11]. Nevertheless, scenarios that consider Ni as the active site have also been proposed[14,15]. In the latter scenario, Fe is assumed to transfer a partial amount of charge to Ni, whereby its oxidation state is stabilized. The possibility of the two cations working concertedly as active sites for different reaction steps has also been suggested[16]. These ambiguous explanations reveal that the true mechanism driving the activity enhancement via Fe doping of NiOOH has remained elusive[3].

[1]Institute of Energy and Climate Research (IEK-13), Forschungszentrum Jülich, Wilhelm-Johnen-Straße, 52425 Jülich, Germany. [2]JARA Energy & Center for Simulation and Data Science (CSD), 52425 Jülich, Germany. [3]Chair of Theory and Computation of Energy Materials, Faculty of Georesources and Materials Engineering, RWTH Aachen University, 52062 Aachen, Germany. ✉e-mail: p.kowalski@fz-juelich.de

Atomistic modeling is nowadays routinely employed to investigate the intrinsic electronic, magnetic, thermal, mechanical etc. properties and the phase behavior of materials, and to study the energetics and kinetics of surface processes in catalysis and electrocatalysis[17,18]. Whereas the crystalline structure, phase behavior, and the electronic as well as ionic properties of $Ni(OH)_2$ are well understood[19], the structure and properties of NiOOH are not conclusively defined[20,21]. Different variants of NiOOH structures could be realized that exhibit: (1) different stacking patterns, (2) different configurations of the cation layer, e.g., in the form of NiO, $Ni(OH)_2$, or NiOOH, and (3) different bond lengths and distribution patterns of hydrogen atoms between the cation layers[22,23]. Some of these patterns are illustrated in Fig. S2. Using simulations based on density functional theory (DFT) and a genetic algorithm, Li and Selloni identified two types of stable structures, consistent with the mosaic texture seen in the transmission electron microscopy (TEM) data[24]. Tkalych et al. found that the staggered $\beta$-NiOOH structure with $Ni^{3+}$ in the (LS) state and an anti-ferromagnetic (AFM) spin arrangement fits best the measured lattice parameters[25]. Conesa investigated 16 plausible structures of $\beta$-NiOOH with different stackings[22], and identified the $3R_C$ structure as the most stable[22]. Martirez et al. proposed two new structures, which they labeled MC1, MC2[23]. Friebel et al. investigated de-hydrated structures, and they suggested hydrated $\gamma$-NiOOH as the OER active phase[9]. They used the DFT+$U$ method to rationalize the "peculiar" similarity of Ni-O and Fe-O bond lengths measured by the extended X-ray absorption fine structure (EXAFS) technique.

Standard DFT-based simulations using the Generalized Gradient Approximation (GGA) functionals incorrectly predict NiOOH compounds to be metals, see Table 1. The DFT+$U$, hybrid functionals, and $G_0W_0$ methods were therefore applied to improve the description of $d$-electron correlations, electronic structure, and band gaps of $Ni(OH)_2$ and NiOOH compounds[22,24,26,27]. Interestingly, the widely used DFT+$U$ method[25,26] also fails to predict the band gap of $\beta$-NiOOH, which measured value lies in the range of $1.7-3.8$ eV[28-30].

The oxidation states of Ni and Fe in NiOOH materials were investigated by atomistic simulations. Although +3 is typically considered the oxidation state of Ni in NiOOH compounds, some studies report the formation of a pair of $Ni^{2+}$ and $Ni^{4+}$[22]. Goldsmith et al. investigated the change of Fe and Ni oxidation states in a single layer of Fe-doped nickel hydroxide, with different levels of deprotonation[27]. They found that a variety of oxidation states of Ni (+2, +3, +4) and Fe (+2, +3, +4, +5) can co-exist, depending on the number of hydrogen atoms in the structures.

The spin state of active surface atoms plays an important role in electrochemical reactions[31,32]. Since all metal cations in the NiOOH lattice have octahedral coordination, crystal field theory predicts the $d$ orbitals of cations to split into two groups: $t_{2g}$ (3 orbitals) and $e_g$ (2 orbitals), with the $e_g$ group being higher in energy. The energy difference between the two groups is the so-called "splitting energy". $Fe^{3+}$ has 5 electrons in the $d$ orbital. If the splitting energy is small, $Fe^{3+}$ prefers a "high-spin" (HS) electronic configuration: $(t_{2g})^3(e_g)^2$; if the splitting energy is large, $Fe^{3+}$ prefers a "low-spin" (LS) electronic configuration: $(t_{2g})^5(e_g)^0$. These configurations correspond to different electronic structures, hence resulting in different performances of the cation as an active site. In order to unravel the role of Fe in enhancing the OER activity of NiOOH, identifying the correct spin state of Fe may be a crucial aspect. The majority of computational studies of Fe-doped NiOOH phases consistently report LS $Ni^{3+}$ and HS $Fe^{3+}$ states[27,33,34]. To the best of our knowledge, the possibility of other spin arrangements of Fe has not been explicitly investigated, nor has the spin state of Fe in NiOOH been measured.

Here, we apply the state-of-the-art DFT+$U$ approach and concepts of thermodynamic modeling of solid solutions to study the properties of Fe-doped NiOOH materials, with the aim to unravel the electronic structure and mixing capability of Fe in these compounds. In particular, we focus on understanding the role of the spin state and the solubility limit of Fe on the OER activity, aspects that have received scarce attention in previous studies.

## Results and discussion

The following analysis is based on the $\beta$-NiOOH and $\beta$-$Ni(OH)_2$ structures. The models of water-containing oxyhydroxide and hydroxide phases are not considered here due to their uncertain structure, complex composition of aqueous interlayers in these materials, and to simplify the analysis. This is in line with previous studies. Friebel et al., for instance, modeled the $\gamma$-NiOOH structure as de-hydrated, with a de-protonated $\beta$-NiOOH model[9].

### Calculation of the electronic structure

The electronic structures of $\beta$-NiOOH and $\beta$-$Ni(OH)_2$ have been simulated in various theoretical works[22-26]. In general, materials with strongly correlated $d$ or $f$ electrons represent a challenge to DFT-based methods and their electronic structure has to be computed with a carefully adapted computational methodology[25,26,35]. The parameter that is widely discussed and serves as a benchmark for computational methods is the band gap. Different studies have reported that the DFT+$U$ method severely underestimates the band gap of $\beta$-NiOOH, resulting in the incorrect prediction that this material is a metal[24-26]. Hybrid functionals have been employed to correct this shortcoming, see Table 1[22,24-26]. The width of the band gap depends on the applied Hubbard $U$ parameter as well as the projectors used for the estimation of $d$ orbital occupancy[36,37]. Here we use $U = 5$ eV for Ni and Fe, which is consistent with previous studies[9,24,27,36]. The choice of $U$ parameter for simulations of Ni is widely discussed in the literature. Similar values to the one used by us are recommended based on the agreement with experimental data, e.g., for magnetic and optical properties, as well as for the band gap[38]. Large $U$ values of $7-8$ eV have also been used[39], but considered as overestimated due to the missing self-screening of $d$ electrons[38,39]. However, values >8 eV are required to correctly predict the lattice parameters and elastic constants of NiO[40]. The suitability of different types of the double-counting correction scheme applied in the DFT+$U$ approach for the computation of transition metals-oxides has also been discussed[41].

### $Ni(OH)_2$ phase

The simulated lattice parameters of the $\beta$-$Ni(OH)_2$ phase agree with the experimental values within 3%, see Table S3. This is a good match, considering the large spread in measured values that result

**Table 1 | The band gaps of $\beta$-$Ni(OH)_2$ and $\beta$-NiOOH calculated with different exchange-correlation functionals**

| Method | $\beta$-$Ni(OH)_2$ | $\beta$-NiOOH |
|---|---|---|
| This work | | |
| DFT | 1.66/0.60 | 2.77/0.00 |
| DFT + $U$ | 2.60/2.55 | 2.61/0.00 |
| DFT + $U$(WF) | 3.72/3.42 | 3.89/3.55 |
| HSE06 | 3.45/3.40 | 2.78/1.1 |
| Previous results | | |
| exp. | 3.6–3.9[a] | 1.7–3.75[a,b,c] |
| DFT | 0.95–2.90[d,e] | 0.01–0.04[f] |
| DFT + $U$ | –/2.98[g] | 0.00–0.19[f,g,h] |
| PBE0 | 3.17/3.17[i] | |
| PBE0$\alpha$ | | 1.00–2.75[*,f,h,i,j] |
| HSE06 | 4.2[k] | 0–1.73[f,h,k] |

If two numbers are reported, these represent direct and indirect band gaps, respectively. The unit is eV.

References: [a]: Ref. 28, [b]: Ref. 30, [c]: Ref. 29, [d]: Ref. 69, [e]: Ref. 70, [f]: Ref. 26, [g]: Ref. 25, [h]: Ref. 44, [i]: Ref. 27, [j]: Ref. 22, [k]: Ref. 45.

*Different values calculated with different amount of the exact exchange, $\alpha$.

from different stacking faults in the samples prepared by different methods[42].

Regarding the electronic structure, nickel hydroxide is an insulator, with a measured band gap of ~3.6–3.9 eV[28]. The band gaps of $\beta$-Ni(OH)$_2$ phases computed using different DFT-based approaches are given in Table 1. The standard DFT+$U$ method results in a smaller band gap (~2.6 eV). We note, however, that Tkalych et al.[25] obtained a much larger band gap of ~3 eV with the DFT+$U$ approach. A wider band gap of 3.17 eV was also obtained with the PBE0 hybrid functional[22,27]. One problem, often overlooked when performing DFT+$U$ calculations, is the selection of correct projectors for the estimation of $d$ orbital occupation. It is well known that atomic orbitals applied in standard DFT+$U$ calculations result in significant fractional occupations of empty $d$ states and an overestimation of the total number of $d$ electrons[35,37,43]. Our DFT+$U$ calculations show a similar behavior, with the total occupation of $d$ orbitals of 8.4, which exceeds the expected value of 8, as shown in Table 2. This deficiency can be corrected by using Wannier-type projectors, hereafter denoted as the DFT+$U$(WF)

method[35,37]. Usage of these projectors results in correct total occupation of ~8.0 (Table 2). This has a significant impact on the derived electronic state. The computed band gap of 3.3–3.7 eV, cf. Table 1, agrees well with the aforementioned experimental measurements. The resulting density of states (DOS) of $\beta$-Ni(OH)$_2$ is shown in Fig. 1. The overall shape resembles closely the DOS functions obtained here with the hybrid functionals (see Fig. 1e), and in studies by Zaffran et al.[26] and Li and Selloni[44]. The selection of proper projectors is thus crucial for the estimation of the occupancy of $d$ orbitals, as required by the DFT+$U$ scheme, and essential for the correct prediction of electronic structure.

## NiOOH phases

NiOOH has been extensively studied with experimental[20,21,28,29] and theoretical[22–26] approaches. The $\beta$-NiOOH phase is usually considered, but with several different structural arrangements. Some of these are illustrated in Figure S2. In Table S2 we provide the lattice parameters for all structural arrangements considered here, and compare these to previous computational studies and available experimental data.

Calculated energies of these structures are provided in Table S1. We identified the MC1 phase to be the most stable configuration, however, with the MC2, TC, FB and EE structures being close in energy, with a difference of just a few kJ/mol per NiOOH unit. This implies the existence of phase mixtures in a real material and explains problems encountered with the XRD-based structure refinement[20]. We note that the experimental study indicates the CP phase as the most stable[20]. However, the measured lattice parameters and atomic structure cannot be reproduced by simulations (see CP(tw) in Table S2).

The measured band gap of $\beta$-NiOOH is ~1.7 eV[28] or ~3.6 eV[29,30], with the larger value reported in more recent studies. The DFT and standard DFT+$U$ methods predict this compound to be a metal. This deficiency is corrected with the DFT+$U$(WF) approach. This method gives correct

**Table 2 | Occupations of $d$ orbitals computed with the DFT and DFT+$U$ approaches, with the atomic orbitals as projectors, and with the DFT+$U$(WF) approach that uses a Wannier functions-based representation as projectors of the $d$ orbitals**

|  | $\beta$-Ni(OH)$_2$ | $\beta$-NiOOH | Fe-doped $\beta$-NiOOH | |
| --- | --- | --- | --- | --- |
|  | Ni$^{2+}$ | Ni$^{3+}$ | Fe$^{3+}_{HS}$ | Fe$^{3+}_{LS}$ |
| DFT | 8.4 | 8.6 | 6.3 | 7.0 |
| DFT+$U$ | 8.4 | 8.5 | 6.2 | 6.6 |
| DFT+$U$(WF) | 8.0 | 6.9 | 4.8 | 5.0 |
| Expected | 8.0 | 7.0 | 5.0 | 5.0 |

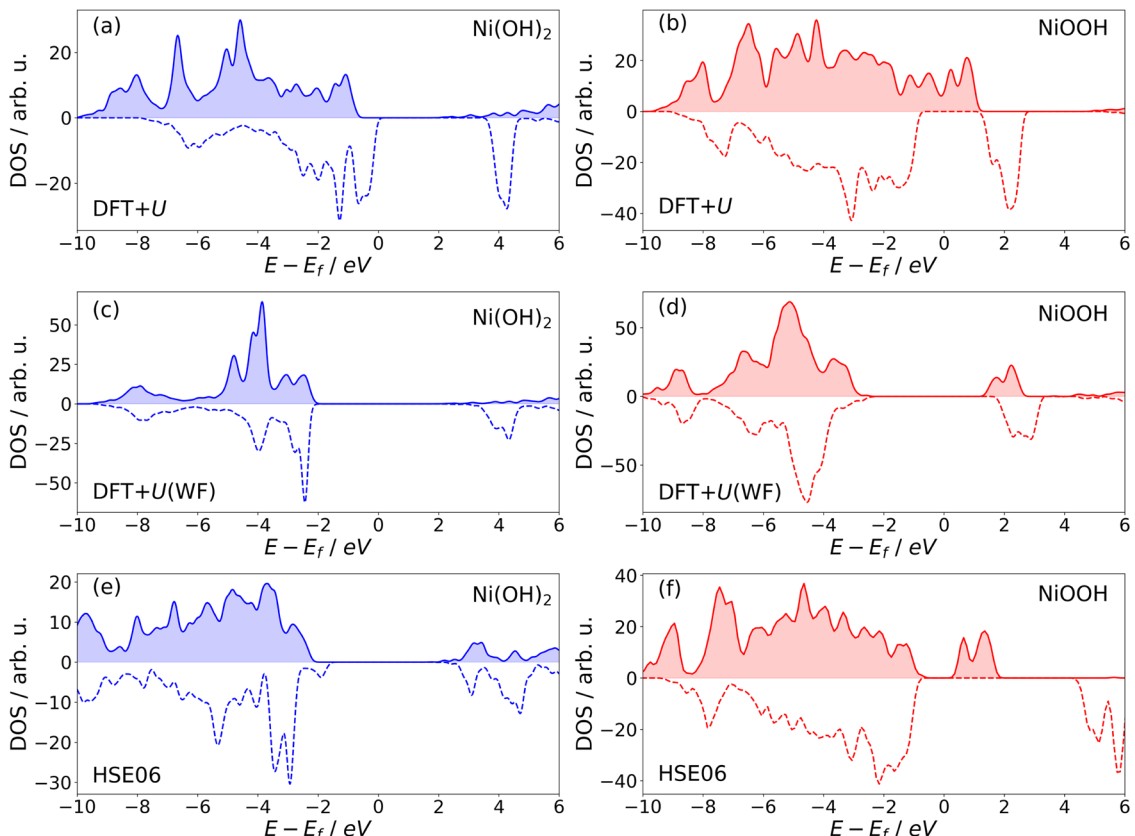

**Fig. 1 | The density of states (DOS) of $\beta$-Ni(OH)$_2$ (left panel, blue) and $\beta$-NiOOH (right panel, red).** The data computed with the DFT+$U$ (**a, b**) and DFT+$U$(WF) (**c, d**) methods and the HSE06 hybrid functional (**e, f**), as indicated. The upper (filled) and lower (non-filled) regions represent the DOS of majority and minority spin states, respectively.

total occupations of $d$ orbitals (expected ~7.0 vs. 8.5 predicted with the atomic orbitals, Table 2). The computed band gaps (3.9 (direct)/ 3.5 (indirect) eV, Table 1) agree with the aforementioned results of more recent experimental studies. The existence of a band gap is also predicted with hybrid functional calculations. Our calculation using the HSE06 functional produced a band gap of 1.1 eV; the results of other studies are also reported in Table 1. The resulting DOS are shown in Fig. 1. As in the case of Ni(OH)$_2$, the DOS obtained with the DFT+$U$(WF) scheme resembles the one obtained with the hybrid functionals (this study and ref. 45).

## Fe-doped $\beta$-NiOOH

In experiment, a maximum in the OER activity was found to coincide with the Fe solubility limit of ~30%[9]. The solubility limit in solid solutions is usually determined by thermodynamic parameters such as the excess enthalpy of mixing and associated Margules interaction parameters[13,46]. These parameters can be derived from atomistic simulations, see e.g.[47], or estimated from the ionic radii of the cations being mixed[13,48]. Ideal mixing is expected to occur in a solid solution when cations are of similar size[47]. Immiscibility, on the other hand, is usually associated with a large mismatch of cation sizes. These rules resemble the Hume-Rothery rules for metallic solid solutions[49]. The size of the host matrix cation determines the configuration formed by the dopant cation embedded within the host lattice, including its resulting oxidation or magnetic state[50-52]. It is known that under high pressure, Fe in ferropericlase or silicate perovskite undergoes a spin transition from HS to LS state[51,52]. The ability of a transition metal cation to exist in LS or HS state also depends on the splitting energy between the $t_{2g}$ and $e_g$ components of $d$ states. The 6-fold coordinated Fe and Ni can exist in both spin states[53].

Because Fe in FeOOH phases exists in the HS state, it is generally assumed that this is also the case for Fe impurities in NiOOH phases[9]. Most ab initio studies and all DFT+$U$-based investigations have found or discussed the HS state for Fe[16,27,34,54,55]. Only one study performed with a hybrid functional computed Fe as LS species in Fe-doped NiOOH[22]. However, this possibility was neither justified nor were its implications ever discussed. Looking at the sizes of cations reported in Table 3, it is striking that the Fe species most similar in size to LS Ni$^{3+}$ is LS Fe$^{3+}$, while the ionic radius of HS Fe is larger by 0.1 Å. This indicates that NiOOH may preferentially incorporate Fe$^{3+}$ as the LS species. If this is true, a spin transition from low to high Fe spin state should occur upon increasing the Fe content in NiOOH. This spin transition may determine the solubility limit of Fe in NiOOH, as the resulting increase in the size mismatch of mixing cations (HS Fe vs. LS Ni) could trigger a phase separation.

Further experimental evidence shows that, while large concentrations of Fe can exist in the NiOOH phase, Ni is not detected as dissolved in the FeOOH phase[9]. At the same time, the measured pure NiOOH and FeOOH phases are structurally different, with $\gamma$-NiOOH phase and lepidocrocite phase of FeOOH ($\gamma$-FeOOH). We note that although the most stable phase of FeOOH is $\alpha$-FeOOH (geothite), $\gamma$-FeOOH forms under aqueous conditions[9]. At standard conditions, the measured enthalpy of formation of $\gamma$-FeOOH is ~10 kJ/mol higher than that of geothite[56]. Solid solutions with different phases of endmembers are often characterized by a wide miscibility gap that is especially

pronounced under ambient conditions[13]. Large solubilities of solute species require much higher temperatures, as observed, for instance, in the monazite-xenotime system, where solubilities larger than 20% are detected at temperatures higher than 1000 °C[13,48].

The solubility limit of Fe in NiOOH and Ni in FeOOH, $x_{Fe}$, and $x_{Ni}$, respectively, can be estimated by numerically solving two coupled non-linear equations, with the interaction energy of Fe and Ni, $W_{Fe}$, and $W_{Ni}$, in the solvent host matrices as parameters (Margules interaction parameters)[13],

$$W_{Fe}(1 - x_{Fe})^2 + RT \ln \frac{x_{Fe}}{1 - x_{Ni}} = -\Delta E(FeOOH) + W_{Ni}x_{Ni}^2, \quad (1)$$

$$W_{Ni}(1 - x_{Ni})^2 + RT \ln \frac{x_{Ni}}{1 - x_{Fe}} = -\Delta E(NiOOH) + W_{Fe}x_{Fe}^2, \quad (2)$$

where $\Delta E$ is the energy (enthalpy) difference between the pure solute phase and a solute with a solvent phase structure. For Fe as a solute, this corresponds to the energy difference, $\Delta E(FeOOH)$, between FeOOH in $\beta$-NiOOH and $\gamma$-FeOOH (lepidocrocite) phase.

Assuming a very low solubility limit of Ni in FeOOH, i.e., $x_{Ni}$ ~ 0, these two equations reduce to[13],

$$W_{Fe}(1 - x_{Fe})^2 + RT \ln x_{Fe} = -\Delta E(FeOOH). \quad (3)$$

The interaction energy of dopant, $W_M$ ($M$ = Fe, Ni), can be estimated from the sizes of the mixing cations using Eq. 7 from Ref. 13,

$$W_M = 4\pi N_a Y \left( \frac{R_h(R_d - R_h)^2}{2} + \frac{(R_d - R_h)^3}{3} \right) + \Delta E(MOOH) = W_0 + \Delta E(MOOH), \quad (4)$$

where $N_a$ is the Avogadro constant, $Y$ is the Young's modulus of the host matrix cation, $R_h$ is the ionic radius of the host matrix and $R_d$ is the ionic radius of the dopant cation. The first term on the right hand side, $W_0$, is the energy needed to incorporate a solute cation into a solvent host matrix, and is estimated as the elastic energy associated with the stress and strain resulting from the mismatch in sizes of solute and solvent cations[13,48]. Because Young's modulus of NiOOH is unknown, we computed it for $\beta$-NiOOH using the standard stress-strain relationship within the Voigt-Reuss-Hill approximation, as applied in previous studies[57]. The obtained value is 104.2 GPa. The resulting Margules interaction parameters are listed in Table 3. In principle, the $W$ energy cannot be larger than a few kJ/mol, a value comparable to the thermal energy at ambient condition (~2.5 kJ/mol), for a homogeneous solid solution to form[46]. The experimental observation that substantial amounts of Fe can enter the NiOOH phase indicates that pure FeOOH with the structure of the NiOOH phase must have an energy very similar to that of the lepidocrocite ($\gamma$-FeOOH).

Our calculations show that $\Delta E$ between the FeOOH phases is ~2 kJ/mol (Table 4). For the LS Fe species this results in similar interaction energy, because of a negligible $W_0$ contribution (Eq. (4)). However, if Fe is incorporated in the HS state, the total interaction energy is too large, ~5 kJ/mol, for Fe to be significantly soluble in NiOOH. The computed solubility limit with this value of the interaction

**Table 3 | The Shannon ionic radii of six-fold coordinated Ni$^{3+}$ and Fe$^{3+}$ species in high and low-spin states, the Magules interaction parameters $W$ from Eq. (4), and the Ni-O and Fe-O bond lengths derived assuming radius of oxygen of 1.38 Å, and computed here with the DFT+$U$(WF) method considering the average of the four shortest bond lengths (in parentheses)**

| | Ni$^{3+}$ | Fe$^{3+}$ | $W_O$ (kJ/mol) | $W_{Fe}$ (kJ/mol) | $W_{Ni}$ (kJ/mol) | Ni-O (Å) | Fe-O (Å) | Ni-O (Å)[9] | Fe-O (Å)[9] |
|---|---|---|---|---|---|---|---|---|---|
| Low-spin | 0.56 | 0.55 | 0.02 | 2.12 | 14.9 | 1.92 (1.92) | 1.91 (1.90) | 1.89 ± 0.02 | 1.91 ± 0.02 |
| High-spin | 0.60 | 0.65 | 2.82 | 4.92 | 17.52 | 1.96 | 2.01 (2.00) | | |

Ionic radii data have been taken from ref. 53. The last two columns report the Ni-O and Fe-O bond lengths measured with EXAFS spectroscopy for 25% Fe concentration in $\gamma$-NiOOH[9].

**Table 4 | The energy differences between the different pure phases of NiOOH and FeOOH**

|  | ΔH (kJ/mol) |
|---|---|
| E(NiOOH in γ-FeOOH form) - E(β-NiOOH) | 14.7 |
| E(FeOOH in β-NiOOH form) - E(γ-FeOOH) | 2.1 |

**Table 5 | The energy differences between systems with Fe in HS and LS state, computed per Fe cation (for a supercell representing an Fe concentration of 6.25 %), in Fe-doped β-NiOOH (denoted as Fe:NiOOH) and γ-FeOOH phases**

|  | E(LS) − E(HS) (kJ/mol) |
|---|---|
| Fe:NiOOH (DFT) | −40.0 |
| Fe:NiOOH (DFT+$U$) | 70.4 |
| Fe:NiOOH (DFT+$U$(WF)) | −39.4 |
| γ-FeOOH | 15.6 |

All the calculations were performed with the PBEsol exchange-correlation functional.

energy is 3%. In case of the Fe LS state, the elastic energy term vanishes and the interaction energy is given by $\Delta E$. It is thus the LS state of Fe and the small energy difference between FeOOH in NiOOH and γ-FeOOH phases that gives rise to the high solubility of ~30% for Fe in NiOOH. On the other hand, $\Delta E$ between NiOOH phases is much larger, ~15 kJ/mol (Table 4), implying that the solubility of Ni in FeOOH should be negligible, with the computed solubility limit of only $2 \times 10^{-4}$ %.

### Low-spin Fe solution

To further support our arguments for the Fe LS species in Fe-doped NiOOH, we performed a series of DFT-based calculations. As seen in previous studies[9,22,27], the standard DFT+$U$ simulations predicted the HS state of Fe as the most stable one, with the energy difference between the two spin states being ~70.4 kJ/mol (Table 5). When the DFT+$U$(WF) method is applied, the solid solution with Fe in the LS state is more stable than that with Fe in the HS state by ~39.4 kJ/mol (see Table 5). This result is consistent with the outcome of our standard DFT calculations (Table. 5) and previous studies performed with an accurate, hybrid functional[22]. It reveals that the widely considered HS state of Fe is an artifact of DFT+$U$ calculations that utilize inadequate projectors.

The LS Fe scenario should be detectable in experiments, by comparing the bond length of Fe-O to that of Ni-O. Because low-spin $Fe^{3+}$ and $Ni^{3+}$ have nearly identical size, the respective bond lengths to oxygen atoms should also be similar. With HS $Fe^{3+}$ solution, the Fe-O bond lengths should be longer by ~0.1 Å. Freibel et al. discussed the "puzzling" similarity of the Fe-O and Ni-O bond lengths in the γ-NiOOH phase, obtained from measurements of the extended X-ray absorption fine structure (EXAFS)[9]. Their finding is consistent with the LS Fe scenario. The estimation of Ni-O and Fe-O bond lengths from the ionic radii of species in different spin states are reported in Table 3. When Ni and Fe are in the LS state, their bond length with oxygen atoms differs only by 0.01 Å. The estimated values match well the measurements of Freibel et al.[9]. This result represents strong evidence in favor of the LS Fe scenario.

### Spin transition of Fe in β-NiOOH

Since Fe exists in the LS state when immersed into a Ni-rich oxyhydroxide phase, and in the HS state in the FeOOH phase, a spin transition is expected to occur at an intermediate Fe concentration. In Fig. 2, we show the computed energy difference between Fe-doped NiOOH phases with HS and LS Fe, as a function of Fe concentration. The computed cross-over point is determined at ~25%, which is consistent with the measured Fe solubility limit (reported as 25% in refs. 9,10, 30% in ref. 58, and 35% in ref. 59). This observation leads us to conclude that the solubility limit of Fe in NiOOH is driven by the spin transition of Fe, which affects the thermodynamics of the solid solution, and causes the opening of a miscibility gap, as discussed in the section "Fe-doped β-NiOOH".

### Fe spin state and OER activity

It is well known that a transition metal dopant can boost the activity of an electrocatalyst. For instance, doping of NiFe-LDH (Layered Double Hydroxide) with V atoms increases the OER activity of that compound. This effect arises because of the occurence of V $d$-band states near the Fermi level[60]. Here we have analyzed the Fe-doped NiOOH material for the observation of the same phenomenon. The DOS obtained for Fe in LS and HS configurations, for the bulk and surface, are shown in Fig. 3,

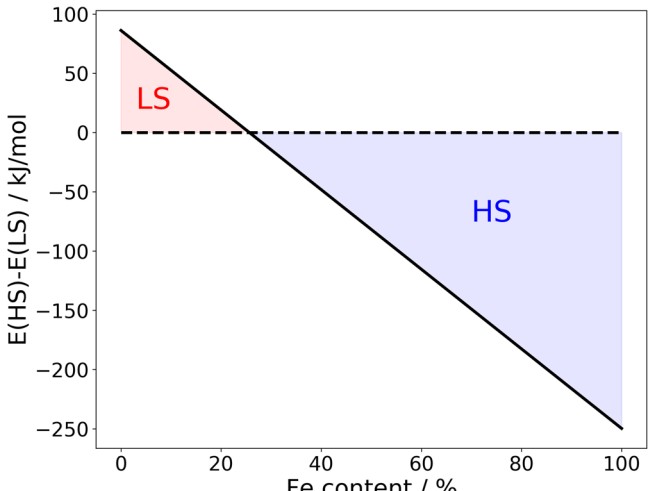

**Fig. 2 | The energy difference between Fe-doped β-NiOOH with Fe HS and Fe LS species, as a function of Fe content.** The red and blue shadowing indicates stability of Fe LS and Fe HS configurations, respectively.

and positions of the Ni and Fe $d$ band centers are reported in Table 6. Interestingly, for the HS state of Fe, the $d$ band center of Fe is shifted by ~0.6 eV farther from the Fermi level than the $d$ band center of Ni. In this case no increase in materials activity is expected with doping. On the other hand, for Fe in the LS state, the $d$ band center of Fe is located closer by ~0.3 eV to the Fermi level than the $d$ band center of Ni, which should exert a positive impact on the activity. This qualitative argument emphasizes the role of the LS state of Fe as the driver of the drastically enhanced activity of Fe-doped NiOOH.

In order to test the impact of the LS state of Fe on OER activity, we computed the reaction energy and overpotential of the potential-determining step (PDS) of the reaction on the Fe-doped NiOOH(0001) surface model. Rajan et al. considered two OER pathways on two distinct active sites on that facet, on O-site (R1) and OH-site (R2)[61]. The two key reaction steps in both scenarios are:

- R1 pathway:

$$*O + H_2O \rightarrow *OOH + (H^+ + e^-)$$
$$*OOH \rightarrow *O_2 + (H^+ + e^-)$$

- R2 pathway:

$$*OH \rightarrow *O + (H^+ + e^-)$$
$$*O + H_2O \rightarrow *OOH + (H^+ + e^-)$$

A previous DFT-based computational study identified the first step of the OER reactions sequence as the PDS[61]. In Table 7, we provide the reaction energies and the resulting overpotentials for the two

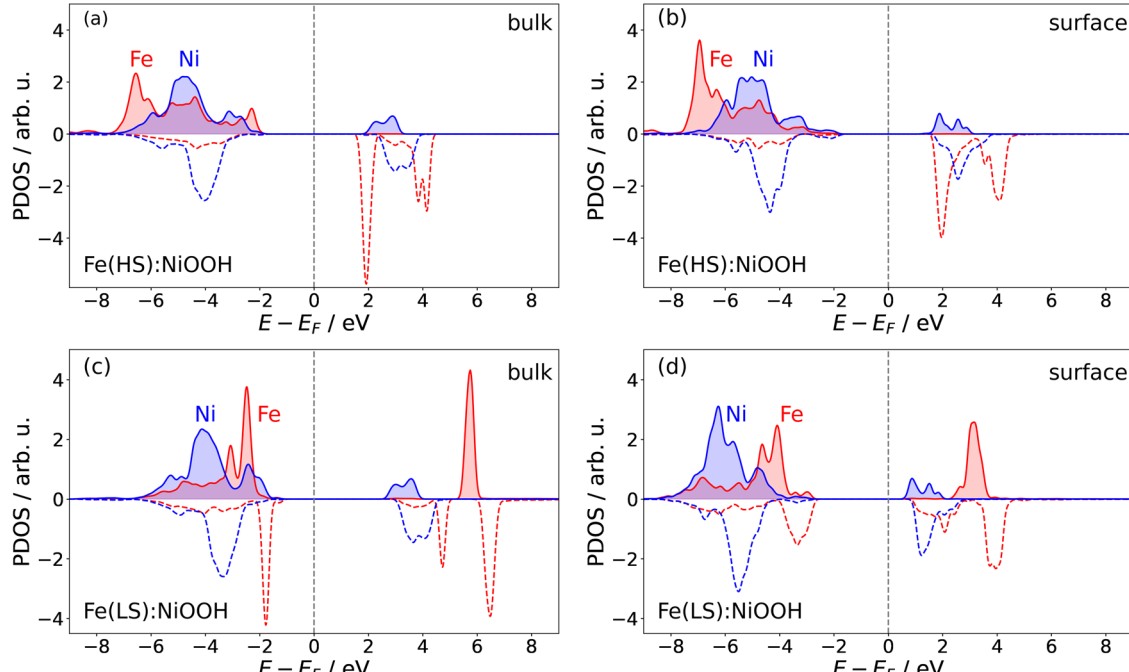

**Fig. 3 | The projected density of states (PDOS).** The PDOS of *d* states of Fe (LS and HS solutions) and Ni cations for Fe-doped NiOOH bulk (left panels (**a**) and (**c**)) and surface (right panels (**b**) and (**d**)). The upper (filled) and lower (non-filled) regions represent the DOS of majority and minority spin states, respectively. The red and blue colors represent *d* states of Fe and Ni, respectively.

considered reaction steps. According to Rajan et al., in the R1 case, Ni changes the oxidation state from +3 to +2[61]. This is also the case with Fe-doped NiOOH, and as a result the free energy of the first step for pure NiOOH and Fe-doped NiOOH is similar. However, Rajan et al. concluded that the R2 pathway is more plausible[61]. In this case, a cation becomes oxidized to the +4 state. Having the *d*-band of Fe in LS state closer to the Fermi level than the *d*-band of Ni, the Fe cation is active surface site for electron donation. Fe is then expected to lose electrons whereas Ni gains electrons. This results in significantly lower reaction-

### Table 6 | The *d* band center ($\epsilon_d$) calculated from DOSes of the surface (Fig. 3)

|  | Ni | Fe |
|---|---|---|
| pure NiOOH | −5.29 |  |
| Fe(HS):NiOOH | −4.80 | −5.37 |
| Fe(LS):NiOOH | −4.69 | −4.36 |
| pure FeOOH(HS) |  | −4.65 |
| pure FeOOH(LS) |  | −5.14 |

The unit is eV.

### Table 7 | The reaction free energies of the first two steps of OER mechanisms R1 and R2 (see the text)

| Reaction pathway | System | step 1 | step 2 |
|---|---|---|---|
| R1 | Ni | **2.67** ($\eta = 1.44$) | 1.74 |
|  | Fe(HS) | **2.38** ($\eta = 1.15$) | 1.75 |
|  | Fe(LS) | 2.62 | 2.09 |
| R2 | Ni | 1.22 | **2.00** ($\eta = 0.77$) |
|  | Fe(HS) | 0.47 | 2.81 |
|  | Fe(LS) | 1.43 | **1.65** ($\eta = 0.42$) |

Ni, Fe(HS), and Fe(LS) mark the values computed for the pure, Fe(HS)- and Fe(LS)-doped β-NiOOH(0001) surfaces, respectively. The number in the brackets represents the overpotentials of OER ($\eta$, in V) determined for the potential-determining step (marked in bold). The unit is eV.

free energy for the second step (Table 7). The LS Fe species should thus, in addition to contributing to the high Fe solubility in NiOOH, accelerate the OER.

Our results indicate, that the PDS for OER is different for the R1 or the R2 pathway (Table 7). For pathway R1, the PDS is the first step, which is in line with the results of Rajan et al.[61]. For pathway R2, the PDS is the second step. We note that this is in line with the theoretical consideration based on the measured potential-dependent Tafel slope[62]. In this case, the LS state of Fe is responsible for the lowest thermodynamic overpotential for the OER. The obtained value, $\eta = 0.42$ V, is significantly lower than the thermodynamic over-potential computed for pure NiOOH compounds, i.e., $\eta = 0.77$ V. These numbers agree well with the experimental values reported by Friebel et al., i.e., 0.36 V and 0.69 V, respectively[9]. This agreement is surprisingly good, given the fact that solvation effects have been neglected in the presented treatment. The computed thermodynamic overpotentials are also not direct equivalents of the kinetic over-potentials. Knowledge of activation barriers and detailed micro-kinetic modeling is required to reveal the true, measurable overpotentials. Nevertheless, it is generally accepted that computed thermodynamic overpotentials provide reasonable approximations of the corresponding kinetic overpotentials. The concept of the thermodynamic overpotential is thus widely used in computational electrochemistry and the thermodynamic overpotentials are often compared to the measured values (e.g.,[61]). Both overpotentials are correlated and exhibit similar trends[63]. For these reasons, we expect a similar decrease in the overpotential, computed or measured, for the OER reaction on the Fe:NiOOH vs. NiOOH electrocatalyst, which we see in the data. For the reaction pathway R2, the sum of the reaction energies of the first and the second step should remain approximately constant, around 3.2 eV, as could also be seen in our results (Table 7). This is because pathway R2 involves the transformation of chemi-sorbed OH to OOH, a process that involves a linear scaling relation between the adsorption energies of OH and OOH, that is given by $\Delta G_{ads}(OOH) = \Delta G_{ads}(OH) + 3.2 \pm 0.2$ eV[64]. Thus, the similar values of the reaction energies of the two steps will result in the lowest possible

overpotential of 0.37 V (computed as $(3.2\,eV - 2 \times 1.23\,eV)/2e$), which is closely approached in the Fe LS scenario.

We note that a real material is more complex in terms of chemical composition, crystal structure, morphology and presence of aqueous phase. However, although we only studied the $\beta$-NiOOH phase, the arguments for the prevalence of the LS state of Fe and for its role in the OER activity should hold for other, more complex NiOOH phases such as $\gamma$-NiOOH.

To summarize our studies, we used theory and atomistic simulation to investigate the incorporation of Fe into NiOOH, and rationalize its role in the enhancement of the OER. We found that, contrary to current understanding, Fe exists in the low-spin state, which is rationalized based on similar sizes of LS $Ni^{3+}$ and $Fe^{3+}$ cations and thermodynamic consideration. The previous assignment of the HS state to Fe resulted from intrinsic difficulties with the standard implementation of the DFT+$U$ method. The use of Wannier-type projectors improved the computed value of the band gap of $Ni(OH)_2$ and NiOOH materials by the DFT+$U$ method. The LS state of $Fe^{3+}$ results in the $d$ band being located much closer to the Fermi level than the $d$ states of Ni. This shift leads to a major enhancement of the OER activity. The thermodynamic considerations of the Fe:Ni solid solution indicate that the LS state of Fe is responsible for the high solubility of Fe in NiOOH. Because in pure FeOOH, Fe exists in the HS state, a spin transition occurs with increasing Fe content. We estimate this transition for the $\beta$-NiOOH phase to occur at 25% Fe content and postulate that it is the spin transition and the related thermodynamics of solid solutions that determine the solubility limit and the OER activity of Fe-doped NiOOH. The computed thermodynamic overpotential of the OER matches well the measured values. Doping with Fe reduces the overpotential of the OER by 0.3 V compared to the undoped NiOOH. The presence of Fe as LS species in NiOOH is thus singled out as the origin of the drastically enhanced OER activity observed in Fe-doped NiOOH.

## Methods

The DFT calculations were performed with the plane-wave Quantum-ESPRESSO package[65], by applying scalar relativistic ultrasoft pseudopotentials to mimic the effect of core electrons[66], and the plane-wave energy cutoff of 80 Ry. Because we were interested in reproducing the structural parameters of the investigated systems, we applied the PBEsol exchange-correlation functional[67]. Calculations were spin polarized, and cell parameters and atomic positions were fully relaxed, reducing the forces acting on the atoms to below 0.001 Ry/$a_0$ (where $a_0$ is the Bohr radius). Different spin states of Fe were computed by constraining the total magnetization of the system. To account for strong correlations between $d$ electrons, we applied the DFT+$U$ method with a Hubbard $U$ parameter of 5 eV for Ni and Fe atoms, and Wannier function-based projectors for counting the occupancy of $d$ orbitals for the DFT+$U$ scheme[37]. The Wannierization procedure was performed with the aid of poormanwannier.x tool, implemented in the Quantum-ESPRESSO code. This method has been successfully applied by us to predict properties of various oxide materials, improving significantly the performance of DFT+$U$ method (e.g., Refs. 35, 37). Monkhorst-Pack approach was used for the $k$-points sampling of the Brillouin zone[68].

The computed structure of $Ni(OH)_2$ is shown in Fig. S1. It was computed with the $2 \times 2 \times 2$ supercell containing 40 atoms on a $5 \times 5 \times 3$ $k$-points grid.

All studied structures of NiOOH are illustrated in Fig. S2. We selected the EE structure as the model of NiOOH for our computational study and analysis. It was computed with the $2 \times 4 \times 2$ supercell containing 64 atoms on the $4 \times 2 \times 2$ $k$-points grid. The $\alpha$-FeOOH and $\gamma$-FeOOH structures (see in Fig. S3) were computed with the $2 \times 4 \times 2$ (64 atoms) supercells on the $4 \times 2 \times 2$ and $5 \times 3 \times 5$ $k$-points grids, respectively.

The surface of $\beta$-NiOOH was modeled with the $2 \times 2$ surface unit cell slab containing 3 layers, adopted from Rajan et al.[61]. The 15 Å thick vacuum layer was applied in the direction perpendicular to the slab surface. The $4 \times 4 \times 1$ $k$-points grid was applied for the sampling of the Brillouin zone.

## Data availability

All the relevant data not included here or in the Supplementary Information are available from the authors upon request.

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

## Acknowledgements

We thank Jun Huang for comments on the OER kinetics and the potential-determining steps, and Andreas Scheinost for information on FeOOH phases. We acknowledge the financial support from Forschungszentrum Jülich GmbH. The presented work was carried out within the framework of the Helmholtz program Materials and Technologies for the Energy Transition under the topic Chemical Energy Carriers and the subtopic Electrochemistry for Hydrogen. The research was funded by the Excellence Initiative of the German federal and state governments and the Jülich Aachen Research Alliance-High-Performance Computing. We thank the JARA-CSD awarding body for time on the RWTH and Forschungszentrum Jülich computing resources awarded through JARA-CSD Partition (Project cjiek61 to P.M.K.). The authors are grateful to the Gauss Centre for Supercomputing e.V. (www.gauss-centre.eu) for providing computing time on the JUWELS Supercomputer at Jülich Supercomputing Centre (Project FZJ-MAC: 26609 to M.J.E.).

## Author contributions

The ab initio calculations and analysis of the results were performed by Z.H., P.M.K., and R.T. P.M.K. performed the thermodynamic modeling and led the project. Z.H. and P.M.K. performed an analysis of the electronic structures, and the OER reaction pathways and edited the manuscript. M.J.E. and M.H.E. contributed to the interpretation of results and the analysis of OER activity of the Fe:NiOOH catalyst. All parties participated in the writing of the manuscript.

## Funding

## Competing interests

The authors declare no competing interests.
