## [Peer Review File · Nature Communications]

REVIEWER COMMENTS

Reviewer #1 (Remarks to the Author):

The DFT+U calculations presented in the paper are adequate. The paper has an argument to support the hypothesis that Iron exists in low spin state in NiOOH. The solubility effect has some indications in the literature also at high spin state modeling at high Fe concentrations, hence the literature should be acknowledged. I recommend publication due to the interesting analysis provided that contributes to the field, subject to revision. Below are some comments for major revision. The key points in the paper are the following:

1. The paper strikes on the conventional fact that the iron exists in high spin state in NiOOH. Authors have proved using DFT+U that the low spin state of Iron in NiOOH is more stable. Earlier DFT calculations predicted that the iron exist in a high spin state. Support by using other functional is recommended.
2. The main idea about the paper is very basic and very simple but has been ignored till now. It is well known that Iron has good solubility upto 30 % in NiOOH. The Ni exists in a Low spin state in NiOOH. If the iron exist in high spin state in the NiOOH then there will be huge difference between the size of the Iron ion and Nickel Ion. This will not lead to such a high solubility of 30% which is seen in the experiments.
3. Authors have shown using the thermodynamic calculations considering the size of the iron in low spin and high spin state as a solute in NiOOH, that high solubility is only achieved when Iron exists in low spin state in NiOOH. If the iron takes high spin state then the calculated solubility is just 3% which is not the case as confirmed by the experiments. I recommend calculating formation energy of the phases.
4. To support their hypothesis, they have experimental evidence from literature which shows that the bond length of Ni-O and Fe-O in NiFeOOH is similar (just a difference of ~ 0.01 Å) which can be only possible if the size of Ni ion and Fe ion is similar. Thus they argue that both exist in the same low spin state. I recommend indicating the distances at both spin states as calculated by DFT+U.
5. Their DFT+U calculations also show that low spin state remains more stable in NiOOH until the iron content exceeds 25 %. Low spin to high spin transition occurs at the iron content around 25 % which is closer to the experimental solubility limit of Fe in NiOOH. The point of solubility effects is actually indicated previously in the literature by Shannon Boettcher's paper on Fe contaminants and referred to in refs. Phys. Chem. Chem. Phys. 19, 7491 (2017); <https://arxiv.org/abs/1902.04788>. I recommend citing these papers.
6. They have also done the calculations for OER on the slab model. They have followed the reaction mechanism different to what reported in the literature. The calculated over voltage with iron in low spin state approach to the experimental value (0.42 V is calculated and 0.36 V is experimentally observed). So here they have made the point that the low spin state of iron in NiOOH is responsible for the high OER activity of the NiFeOOH.

Reviewer #2 (Remarks to the Author):

The manuscript reports that iron existing in a low-spin state in an Fe doping NiOOH OER catalyst plays a crucial role for its high activity using DFT+U calculation method. The work is interesting and meaningful. It could be accepted for publication after addressing some important concerns. Firstly, it lacks direct experimental evidence to convince me. Although the calculation results explained some experimental results reported in literature, the mechanism of oxygen evolution reaction is quite complicated which also depends on the morphology, crystal structure and chemical composition of the catalysts as well as the interfacial structure. The current calculation only considered some simple situations. Secondly, the calculation based on DFT+U is more likely related to the thermodynamic and equilibrium, while the overpotential is more related to the kinetics and non-equilibrium state. The physical meaning of the thermodynamic overpotential is unclear.

Reviewer #1 (Remarks to the Author):

The DFT+U calculations presented in the paper are adequate. The paper has an argument to support the hypothesis that Iron exists in low spin state in NiOOH. The solubility effect has some indications in the literature also at high spin state modeling at high Fe concentrations, hence the literature should be acknowledged. I recommend publication due to the interesting analysis provided that contributes to the field, subject to revision. Below are some comments for major revision. The key points in the paper are the following:

RE: We would like to thank the Reviewer for positive feedback on our paper and useful suggestions. All of them have been implemented.

1. The paper strikes on the conventional fact that the iron exists in high spin state in NiOOH. Authors have proved using DFT+U that the low spin state of Iron in NiOOH is more stable. Earlier DFT calculations predicted that the iron exist in a high spin state. Support by using other functional is recommended.

RE: In our studies we applied the state-of-the-art computational methodology, including the PBEsol exchange-correlation functional and the DFT+U method with Wannier-type projectors. This approach was specifically selected to obtain high accuracy in calculated structural parameters (e.g., Ni-O and Fe-O bond lengths) and electronic structure (e.g., band gap), which are key aspects for the investigation of Fe:NiOOH solid solutions and their OER activity. The low spin state has also been found in independent studies (Conesa, J. Phys. Chem. C 120, 34, 18999–19010 (2016)) that had used an accurate but computationally intensive hybrid functional, although without discussing the implications of the result on structural or catalytic properties. These results are strong support for computational findings reported in our article. We improved the relevant text in Section 2.5.

2. The main idea about the paper is very basic and very simple but has been ignored till now. It is well known that Iron has good solubility upto 30 % in NiOOH. The Ni exists in a Low spin state in NiOOH. If the iron exist in high spin state in the NiOOH then there will be huge difference between the size of the Iron ion and Nickel Ion. This will not lead to such a high solubility of 30% which is seen in the experiments.

RE: This is a correct comment related to one of our key findings.

3. Authors have shown using the thermodynamic calculations considering the size of the iron in low spin and high spin state as a solute in NiOOH, that high solubility is only achieved when Iron exists in low spin state in NiOOH. If the iron takes high spin state then the calculated solubility is just 3% which is not the case as confirmed by the experiments. I recommend calculating formation energy of the phases.

RE: The difference in formation energy of the two phases has been computed and is reported in Table 5 as $E(\text{LS})-E(\text{HS})=-39.4$ kJ/mol per Fe cation, derived at Fe concentration of 6.25%. The text in caption of Table 5 has been improved.

4. To support their hypothesis, they have experimental evidence from literature which shows that the bond length of Ni-O and Fe-O in NiFeOOH is similar (just a difference of ~ 0.01 Å) which can be only possible if the size of Ni ion and Fe ion is similar. Thus they argue that both exist in the same low spin state. I recommend indicating the distances at both spin states as calculated by DFT+U.

RE: We have added the DFT+U bond-lengths to Table 3.

5. Their DFT+U calculations also show that low spin state remains more stable in NiOOH until the iron content exceeds 25 %. Low spin to high spin transition occurs at the iron content around 25 % which is closer to the experimental solubility limit of Fe in NiOOH. The point of solubility effects is actually indicated previously in the literature by Shannon Boettcher's paper on Fe contaminants and referred to in refs. Phys. Chem. Chem. Phys. 19, 7491 (2017); <https://arxiv.org/abs/1902.04788> I recommend citing these papers.

RE: In the introduction, we already cite the paper by Shannon Boettcher. We added a citation to Hamal, & Toroker (2021) on the Fe solubility limit. Other citations had to be omitted because of the limit of 70 citations enforced by the Journal.

6. They have also done the calculations for OER on the slab model. They have followed the reaction mechanism different to what reported in the literature. The calculated over voltage with iron in low spin state approach to the experimental value (0.42 V is calculated and 0.36 V is experimentally observed). So here they have made the point that the low spin state of iron in NiOOH is responsible for the high OER activity of the NiFeOOH.

RE: Correct. This is only a comment by the Reviewer.

Reviewer #2 (Remarks to the Author):

The manuscript reports that iron existing in a low-spin state in an Fe doping NiOOH OER catalyst plays crucial role for its high activity using DFT+U calculation method. The work is interesting and meaningful. It could be accepted for publication after addressing some important concerns.

RE: We thank the Reviewer for the positive feedback and valuable comments, which we have addressed in the revised version.

Firstly, it lacks direct experimental evidence to convince me. Although the calculation results explained some experimental results reported in literature, the mechanism of oxygen evolution reaction is quite complicated which also depending on the morphology, crystal structure and chemical composition of the catalysts as well as the interfacial structure. The current calculation only considered some simple situations.

RE: Obtaining direct evidence of Fe low spin state is by no means a trivial undertaking, but we are certain that the publication of our results will trigger experimental efforts to this effect. We also fully agree that the real system is complex – which however does not diminish the value of our results. Our study focuses on a certain, previously overlooked aspect of the spin state of Fe, and it provides, in our view, strong evidence of its importance on the solubility limit of Fe and on the activity of the OER reaction. Where possible, we demonstrate the agreement with measured structural and electrochemical data. We see this as solid evidence. We added the relevant statement regarding the complexity of the real system at the end of Section 2.7.

Secondly the calculation based on DFT+U is more likely related to the thermodynamic and equilibrium, while the overpotential is more related to the kinetics and non-equilibrium state. The physical meaning of the thermodynamic overpotential is unclear.

RE: Indeed, activation barriers and possibly detailed microkinetic modeling would be required to compute the true kinetic (measured) overpotentials. Nevertheless, it is generally accepted that computed thermodynamic overpotentials provide reasonable approximations of the corresponding kinetic overpotentials; indeed the values are often close (within 0.1 V). The concept of the thermodynamic overpotential is widely used in computational electrochemistry and the thermodynamic overpotentials are often compared to the measured values, including studies of Rajan et al. *J. Am. Chem. Soc.* (2020) that we refer to. Both overpotentials are correlated and exhibit similar trends, as discussed, for instance, in Koper, *J Solid State Electrochem* 17, 339–344 (2013); Man et al., *ChemCatChem* 3, 1159 – 1165 (2011). In Section 2.7 we added relevant clarification.

REVIEWERS' COMMENTS

Reviewer #1 (Remarks to the Author):

There are still a few minor points not fully addressed. In referee 1 report, comment 1: there are no demonstrations of more functionals. There is support with only one literature citation on the existence of low spin state (Conesa et al.), yet there exists a huge number of DFT calculations on NiOOH with spin state reports. Regarding question 3, if the LS is more stable then how was the HS obtained technically? Also, can the formation energy be calculated at more concentrations of Fe doping? It is recommended to report the structure files in SI for reproducibility. I think this would be valuable for the scientific community.

Reviewer #3 (Remarks to the Author):

I am satisfied with the revision and recommend its acceptance for publication in this journal.

Reviewer #4 (Remarks to the Author):

Comments

The manuscript presents a computational study of the Fe-doped NiOOH (Fe:NiOOH) oxygen evolution reaction (OER) electrocatalyst. The authors used DFT+U method with Wannier function-based projectors to demonstrate that at low concentrations (<25%) iron exists in its low-spin state, in contrast to the predictions of the earlier DFT calculations. This conclusion is also supported by the calculations using a more accurate hybrid functional. In addition, the calculations of the thermodynamics of key intermediates in the OER reaction provide theoretical support for the hypothesis that the presence of the Fe in the low-spin state with the Fe *d*-band closer to the Fermi level leads to the dramatic enhancement of the catalytic activity of Fe:NiOOH catalyst.

In overall, the paper is well-written and the conclusions are strongly supported by the calculation data. The main result of the paper about the low-spin state of the Fe in Fe:NiOOH catalyst is a very important research outcome and will be of great interest to the audience of Nature Communications. The current revision of the paper seemed to address all the major concerns of the reviewers. In my opinion, the paper is now suitable for publication subject to minor optional revisions which would address the following minor issues:

- (1) It is unclear if there exists any experimental data (apart from solubility) that would support the low-spin state of iron in Fe:NiOOH. Are there any Mössbauer spectroscopy data available?
- (2) All the calculations in the paper were performed in the gas phase. It is probably adequate for the calculations of the bulk NiOOH and Fe:NiOOH phases but the thermodynamics of the OER intermediates at the surface of the catalyst could be significantly affected by the interactions with the solvent. I think the authors should at least comment on the possible importance of solvation effects.
- (3) [Page 13, Table 5] It is unclear what kind of functional was used in the *standard DFT calculations* (first line of Table 5). Please clarify and use a reference if appropriate (maybe in a footnote).
- (4) [Page 15, Figure 2] I assume that the data plotted in Figure 2 was obtained in DFT+U(WF) calculations. It would be helpful for the reader if it was explicitly stated in the caption.

Reviewer #1 (Remarks to the Author):

There are still a few minor points not fully addressed.

In referee 1 report, comment 1: there are no demonstrations of more functionals. There is support with only one literature citation on the existence of low spin state (Conesa et al.), yet there exists a huge number of DFT calculations on NiOOH with spin state reports.

RE: We believe that, in the previous rebuttal and in the manuscript, we have given extensive explanation on the reasons of failure of previous DFT+U studies, which obtained high spin state of Fe, and justification of our educative choice of computational methodology, including the PBEsol functional applied by us. Nevertheless, we tested with other common DFT functionals (e.g. PBE) and results are consistent with our prediction.

Regarding question 3, if the LS is more stable then how was the HS obtained technically?

RE: Simply, by fixing total magnetization of the system. We added this information to the Methods section.

Also, can the formation energy be calculated at more concentrations of Fe doping?

RE: Such a calculation would be technically more difficult, due to need for accounting for statistical distribution of ions and other factors, and thus results would be questionable. For the diluted concentration of Fe, the single defect method, as applied, should give good estimate.

It is recommended to report the structure files in SI for reproducibility. I think this would be valuable for the scientific community.

RE: Done.

Reviewer #3 (Remarks to the Author):

I am satisfied with the revision and recommend its acceptance for publication in this journal.

RE: We thank the Reviewer for his/her effort and positive evaluation of the manuscript.

Reviewer #4 (Remarks to the Author):

Review attached

The manuscript presents a computational study of the Fe-doped NiOOH (Fe:NiOOH) oxygen evolution reaction (OER) electrocatalyst. The authors used DFT+U method with Wannier function-based projectors to demonstrate that at low concentrations (<25%) iron exists in its low-spin state, in contrast to the predictions of the earlier DFT calculations. This conclusion is also supported by the calculations using a more accurate hybrid functional. In addition, the calculations of the thermodynamics of key

intermediates in the OER reaction provide theoretical support for the hypothesis that the presence of the Fe in the low-spin state with the Fe d-band closer to the Fermi level leads to the dramatic enhancement of the catalytic activity of Fe:NiOOH catalyst. In overall, the paper is well-written and the conclusions are strongly supported by the calculation data. The main result of the paper about the low-spin state of the Fe in Fe:NiOOH catalyst is a very important research outcome and will be of great interest to the audience of Nature Communications. The current revision of the paper seemed to address all the major concerns of the reviewers. In my opinion, the paper is now suitable for publication subject to minor optional revisions which would address the following minor issues:

RE: We thank the Reviewer for very positive reception of our paper and constructive feedback.

(1) It is unclear if there exists any experimental data (apart from solubility) that would support the low-spin state of iron in Fe:NiOOH. Are there any Mössbauer spectroscopy data available?

RE: We are not aware of any Mössbauer spectroscopy or equivalent measurements. We added relevant statement in the introduction.

(2) All the calculations in the paper were performed in the gas phase. It is probably adequate for the calculations of the bulk NiOOH and Fe:NiOOH phases but the thermodynamics of the OER intermediates at the surface of the catalyst could be significantly affected by the interactions with the solvent. I think the authors should at least comment on the possible importance of solvation effects.

RE: The effect of solvation is certainly a topic for a follow-up studies. It could possibly impact the computed reaction free energies and resulting overpotentials, but should not affect the LS solution for Fe. We added a comment on this in section 2.7.

(3) [Page 13, Table 5] It is unclear what kind of functional was used in the standard DFT calculations (first line of Table 5). Please clarify and use a reference if appropriate (maybe in a footnote).

RE: The information has been added.

(4) [Page 15, Figure 2] I assume that the data plotted in Figure 2 was obtained in DFT+U(WF) calculations.

It would be helpful for the reader if it was explicitly stated in the caption.

RE: The information has been added.